# Estimation of the Effects of CO_2_ and Temperature on the Swelling of PS-CO_2_ Mixtures at Supercritical Conditions on Rheological Testing

**DOI:** 10.3390/polym14173490

**Published:** 2022-08-26

**Authors:** César Miguel Ibarra-Garza, Cecilia D. Treviño-Quintanilla, Jaime Bonilla-Ríos

**Affiliations:** 1School of Engineering and Science, Tecnologico de Monterrey, Monterrey 64849, Mexico; 2Institute of Advanced Materials for Sustainable Manufacturing, Tecnologico de Monterrey, Queretaro 76146, Mexico; 3Laboratorio Nacional de Manufactura Aditiva y Digital (MADIT), Autopista al Aeropuerto, Km. 9.5, Calle Alianza Norte 100, Apodaca 66629, Mexico

**Keywords:** polystyrene, carbon dioxide, swelling, viscoelasticity

## Abstract

The use of supercritical CO_2_ as a blowing agent for polymeric foams instead of traditional blowing agents has been a trend in recent years. To achieve the final desired properties of the polymeric foams, the rheological behavior of the material needs to be reliable. The polymer swelling in the samples for rheological testing affects the results of the viscoelastic properties of the material. This study proposes a new testing methodology to control the accuracy and repeatability of the rheological characterization for PS-SCO_2_ samples. To develop this methodology, three polystyrene resins with different molecular weight distribution were studied at three temperatures (170, 185 and 200 °C) and three pressures (0.1 MPa, 6.89 MPa and 13.78 MPa). The CO_2_ concentration was estimated and used in the Sanchez–Lacombe Equation of State (SLEOS) to determine the polymer swelling, as it affects the dimensions of specimens tested in high-pressure rheometers. The correction factors provided a consistent trend in the viscosity with respect to temperature and a decrease of up to 50% in the standard deviation. The results of this study are crucial for an accurate measurement of viscoelastic properties by parallel-plate rheometry.

## 1. Introduction

Polymeric foams are produced by adding a blowing agent into a molten polymer. These foams can be found in airplanes and automotive parts, as acoustic or thermal insulators for construction or appliances and as varieties of cushioning for furniture or packaging to avoid product damage. One of the most important methods for producing polymeric foams is the extrusion process that uses supercritical fluids (SCFs) as blowing agents.

The effect of SCFs has been investigated in a variety of polymers: polyethylene, polypropylene, and polystyrene, among others. Sc-CO_2_ has received special attention, as it is environmentally friendly and can be easily removed by depressurization, reducing, for instance, the cost and processing time compared to other methods that require drying or solvent removal. Another benefit of using Sc-CO_2_ for polymer foaming is the polymers’ plasticization at low temperatures, preventing thermal degradation of resins during processing.

Understanding the mixing of supercritical fluids in a polymer matrix requires accurate measurements of viscoelastic properties under high pressures and temperatures. For high-pressure rheology, different geometries can be adapted for drag and pressure flows. The most-used rheometer consists of a slit-die rheometer first adapted by Han [1]. This configuration was later used by Han and Ma, Royer et al., and Lee et al. Others have used a Couette rheometer [2] and a parallel plate [2,3]. However, available equipment consisting in a parallel-plate rheometer inside a pressure cell incurs errors related to configuration [4,5]. 

It has been proved that for any rheological measurements, the errors can increase up to 7% when testing with samples with a smaller diameter than the plate’s (underfilling) [6] and up to 30% when overfilling happens [4]. Therefore, it is important to know in advance the corrections for volume, thermal expansion and swelling to adjust the specimen dimensions before the rheological test is conducted. Different authors were focused on studying the shrinkage problem in polymer foams with SCFs [7] or characterizing the thermal properties of the foams [8] without developing a method to reduce the measurement error by the rheometers.

This study developed a testing methodology to provide accuracy and repeatability of rheology testing using a parallel-plate rheometer in a pressure cell, supported by the Sanchez–Lacombe Equation of State (SLEOS) [9] The study covers a wide range of operating conditions using temperatures from 170 to 200 °C and pressures of 0.01, 6.89 and 13.78 MPa of CO_2_ for later use in the parallel-plate rheometry. 

## 2. Materials and Methods

### 2.1. Materials

Standard characterization methods were performed for each sample, such as gel permeation chromatography (GPC) for molecular weight distribution (MWD) data from Malvern Instruments, Houston, TX, USA. Also differential scanning calorimetry (DSC) for the glass transition temperature valuation using a DSC from TA Instruments, New Castle, DE, USA

Three polystyrene resins with different molecular weight distribution (MWD) were used in this study (Table 1). These were provided by Total Petrochemicals & Refining USA, Inc., Deer Park, TX, USA.

Here, M_n_ is the number average molecular weight, M_w_ is the average molecular weight, M_z_ is the third moment of the MWD, M_w_/M_n_ is the polydispersity index, and the Peak Mw is the highest peak of the Mw curve. This Peak M_w_ is usually extracted for very narrow distributions of M_w_ in polymers.

### 2.2. Sample Preparation

13.4 and 14.6 mm diameter round samples with 1.5 mm of thickness were fabricated using a hydraulic press, Carver Inc, Wabash, IN, USA., molded at 204 °C in two cycles. First, the resins were stabilized with 2% BHT (butylhydroxytolune) by Sigma-Aldrich, Burlington, MA, USA. The first cycle consisted in 5 min of pressing at 20.7 MPa, followed by a second cycle of 5 min at 413.7 MPa.

### 2.3. Oscillatory Rheometry

High-pressure rheology data were obtained from strain-controlled frequency sweeps data using an Anton Paar SmartPave MCR101, Graz, Austria, rheometer equipped with a pressure cell with a 20 mm parallel-plate configuration and 1 mm gap at 170 °C, 185 °C and 200 °C and two pressures, 6.89 MPa and 13.78 MPa, in addition to atmospheric pressure (0.10 MPa). The strain was fixed at 10%. 

Figure 1 shows the experimental setup consisting of two Teledyne ISCO syringe pumps by Lincoln, NE, USA, with a D-series pump controller. A CO_2_ cylinder is connected to syringe pump B, which is responsible for filling up pump A. Two check valves help the pressurization process for both pumps. After pressurization, a manual gate valve is opened to pressurize the cell (already sealed and loaded with a sample) while pump A is running.

To obtain accurate viscosity data from the rheometer, an experimental methodology was developed. This methodology ensures the repeatability of the tests. As seen in the rheometer’s schematic (Figure 1), the equipment design does not allow manipulation of the sample once placed in testing position. The first step is to prepare the samples and gauge the gap manually to assure the dimensions, and then defining the time of measurements and solubility is required. Under these considerations, the swelling and the actual radius were calculated so that viscosity data could be adjusted. The complete methodology of the experimental process is seen in Figure 2. This methodology’s repeatability is affected by the five elements seen in Figure 2: sample dimensions, gap, sample deformations, degradation and measurement time. This study proposed a setup to reduce the errors in the experimental data controlling these five elements.

### 2.4. Volumetric Thermal Expansion

In order to have a gap control in the measurements, the volumetric thermal expansion of the samples was measured. All the samples were heated and deformed for 10 min at a fixed temperature. Then, the pressure was opened, and the final diameter and thickness were measured using a Vernier caliper.

The volumetric thermal expansion coefficient (*β_T_*) was calculated using Equation (1):(1)$$V_{F}=V_{0}\left(1+\beta_{T}\Delta T\right)$$
where *V*_0_ and *V_F_* are the initial and final volume, respectively, and Δ*T* is the difference between initial (*T*_0_) and final temperature (*T_F_*).

### 2.5. CO_2_ Concentration

In order to use the SLEOS, first, defining the solubility of CO_2_ in PS is required. Henry constant works well for modeling this system; therefore, Equation (2), which was reported by Sato et al. [10], was used for PS + CO_2_:(2)$$\ln\left(k_{P}\right)=6.400+2.537{\left(\frac{T_{c}}{T}\right)}^{2}$$
where *k_P_* is the Henry constant (kg. MPa/cm^3^ STP), *T_C_* is the CO_2_ critical temperature (304.2 K) [11], and T is the temperature of the system. This equation has been proven to correlate experimental Henry constant within average relative deviations of 2.5% for this system [10].

### 2.6. Swelling

Thermodynamic equations of states (EOS) are needed to model the interactions between the polymer and SCFs to describe the system state in terms of pressure and temperature, which will determine solubility, the density of the mixture and even phase stability.

The SLEOS [12] is the most used for polymer-SCF systems, and its effectiveness has been proved for describing different polymer-SCF systems accurately [2,10,13,14,15]. It can be used for calculating the density of a mixture in correlation with the absorption of a gas into the polymer, the fractional free volume, swelling, and phase stability. Therefore, SLEOS is useful in high-pressure rheology because it indirectly considers the plasticization effect of changes in free volume as the gas dissolves in the polymer system.

SLEOS is based on the chemical potential of a system (3) at equilibrium (4), given by:(3)$$\mu =rN\epsilon^{*}\left[-\tilde{\rho }+\tilde{P}\tilde{\upsilon }+\tilde{T}\tilde{\upsilon }\left[\left(1-\tilde{\rho }\right)\ln\left(1-\tilde{\rho }\right)+\frac{\tilde{\rho }}{r}\ln\tilde{\rho }\right]\right]$$
(4)$${\tilde{\rho }}^{2}+\tilde{P}+\tilde{T}\left[\ln\left(1-\tilde{\rho }\right)+\left(1-\frac{1}{r}\right)\tilde{\rho }\right]=0$$
where *N* is the total number of moles; *r* is the number of lattices occupied by a molecule; $\tilde{\rho }$, $\tilde{T}$, $\tilde{\upsilon }$ and $\tilde{P}$ are reduced density, temperature, volume and pressure, respectively, according to their characteristic values (superscript *), and *ϵ** is the characteristic energy of interaction per mer, that is:(5)$$\tilde{T}=T/T^{*}$$
(6)$$\tilde{P}=P/P^{*}$$
(7)$$\tilde{\rho }=\rho /\rho^{*}$$
(8)$$\tilde{\upsilon }=\upsilon /\upsilon^{*}$$

Characteristic parameters for pure components are determined by correlating empirical PVT data with Equation 4. Thus, measurements of solubility and swelling at equilibrium are required. Detailed procedures for determining characteristic parameters can be found elsewhere [9,16]. These procedures can be carried out using experimental setups for the pressure decay method or gravimetric method [17,18].

The SLEOS works with a dimensionless binary interaction parameter (*k*_12_) that considers the deviations caused by pressure, and it is considered in the calculation of the characteristic pressure of the mixture (*P**), which accounts for the cohesiveness of the fluid [12]. According to findings from Sato et al. [19,20], the binary interaction parameter can be modeled as temperature-dependent. Therefore, values from Sato et al. were used to predict the binary interaction parameter for the desired temperatures via a linear regression, resulting in Equation (9):(9)$$k_{12}=-0.0011T+0.330$$

Other values for the interaction parameter can be found elsewhere [2,19].

### 2.7. CO_2_ and PS Properties

The density for both the PS and CO_2_ was estimated according to the following models.

#### 2.7.1. Carbon Dioxide Properties

Density for CO_2_ at supercritical state was calculated using Peng–Robinson (PR) EOS [21]:(10)$$P=\frac{R_{g}T}{V-b_{PR}}-\frac{a\left(T\right)}{V\left(V+b_{PR}\right)+b_{PR}\left(V-b_{PR}\right)}$$
where:(11)$$a\left(T\right)=\frac{0.45724R_{g}^{2}T_{c}^{2}}{P_{c}}*\alpha \left(T\right)$$
(12)$$b_{PR}=0.07780R_{g}T_{c}/P_{c}$$
(13)$$\alpha \left(T\right)={\left[1+k\left(1-T_{r}^{0.5}\right)\right]}^{2}$$
(14)$$k=0.3746+1.54226\omega_{a}-0.26992\omega_{a}^{2}$$
where *P* is pressure, *R_G_* is the gas constant, *T* is temperature, *a*, *b_PR_* and *k* are the Peng–Robinson parameters, *V* is volume, *T_c_* is the critical temperature, *P_c_* is the critical pressure, $\omega_{a}$ is the acentric factor, and $T_{r}$ is the reduced temperature.

#### 2.7.2. Polystyrene Properties

For PS, there are no equations to exactly calculate the density; however, there are some models reported by Mark [22] that show agreement with experimental data within a range of operating conditions. Equation (15) can estimate the PS density for different temperatures at atmospheric pressure:(15)$$\rho =a_{0}+a_{1}T+a_{2}T^{2}$$
where $a_{0},\;a_{1}$ and $a_{2}$ are constants that can be found in Table 2, depending on the temperature range and the reference by which this information was obtained.

On the other hand, to calculate densities as a function of temperature and pressure, the following model is reported [22]:(16)$$\rho \left(P,T\right)=\frac{\rho \left(0,T\right)}{1-C\cdot \ln\left(1+\frac{P}{B\left(T\right)}\right)}$$
(17)$$B\left(T\right)=b_{0}e^{-b_{1}T}$$
where $\rho \left(0,\;T\right)$ represents the density at atmospheric pressure and reference temperature, and $b_{0}$ and $b_{1}\;$ are constants shown in Table 3, at desired operating conditions.

## 3. Results and Discussion 

Since the presence of CO_2_ changes the material properties and dimensions, the amount of CO_2_ dissolved into the polymer and the resulting swelling are needed at each testing condition for the testing to be accurate. In addition, thermal expansion must be considered to validate testing results, as samples’ dimensions are affected by CO_2_ concentration and temperature.

### 3.1. Volumetric Thermal Expansion

Polymer samples need to cover the rheometer’s plate completely to avoid errors due to underfilling or excess on the edges. The results of the calculated volumetric expansion using Equation (1) are shown in Table 4.

Here, *T*_0_ is initial temperature, *T*_F_ is the final temperature, *V*_0_ is the initial volume of the samples, *V_F_* is the final volume of the samples, and *β_T_* is the volumetric thermal expansion coefficient.

An average value of 1.8 × 10^−3^/°C was used as the volumetric thermal expansion coefficient. Validation of this empirical value was performed with data from the literature. Averaged volumetric thermal expansion coefficients showed agreement with data reported by Mark [22] 

### 3.2. Solubility of CO_2_


Using the Henry constant, the solubility (g CO_2_/g PS) for each pair of operating conditions was calculated as seen in Table 5.

The predicted values for the binary interaction (*k*_12_) parameters for PS and CO_2_ mixtures obtained by linear regression (Equation (9)) are shown in Table 6. 

To calculate the mixture properties of PS + CO_2_ systems, some characteristic parameters are needed. These have been reported in different studies [2,12,13,25], but the ones used for this research are presented in Table 7, reported by Sato et al. [10].

Other inputs for SLEOS, in addition to operating temperature and pressure, are the material’s density and molecular weight. For PS, the weight-averaged molecular weight (M_w_) was taken as the molecular weight.

### 3.3. Materials Properties Estimation

#### 3.3.1. Carbon Dioxide

Table 8 and Table 9 present the calculated critical properties and densities of CO_2_ using the Peng-Robinson (PR) EOS (Equation (10)) at different temperatures and pressures.

#### 3.3.2. Polystyrene

It can be seen from Table 2 and Table 3 that the temperature and pressure ranges in which both sets of constants work are compatible to the ones used in this study; therefore, averages of the results from both sets were calculated and are shown in Table 10.

It is important to mention that it was not possible to measure the real density of each resin due to lack of appropriate equipment, and the values presented in Table 10 were considered the same for the three resins.

### 3.4. Swelling

Having the solubility, the interaction parameter and the materials properties, it was possible to use the SLEOS to estimate the swelling. The estimated volumes of the mixture using SLEOS (Equations (3)–(9)) are shown in Table 11. The expected behavior, namely that the higher the temperature and pressure, the higher the CO_2_ solubility and, therefore, the higher the swelling, can be verified through the results obtained by SLEOS.

The assumption of a fixed gap of 1 mm impacts the volume and therefore the radius sample. The volume of a cylinder (sample’s shape) is defined by:(18)$$V_{cl}=\pi R^{2}H$$
where *R* is the sample’s radius and *H* is the height, in this case equal to or defined by the gap. 

In any case, the volume increases due to the thermal expansion and swelling caused by the dissolution of CO_2_ in the PS specimen. It has been shown that, on average, the effect for thermal expansion is 87% while the rest is due to swelling when at 6.89 MPa, and the effect is 77% when at 13.78 MPa. Therefore, the effect on the specimen is around 10%, which could affect the results by approximately 52%.

Finally, having estimated the swelling ratio via SLEOS, the real volume can be used to calculate the final sample’s radius. Based on this experiment and considering that the thickness is fixed by the plates (gap), the final discs dimensions for each temperature are shown in Table 12. 

Here, T is temperature, d_i_ is the initial diameter, h_i_ is the initial height, β_T_ is the volumetric thermal expansion coefficient, H is the gap, and d_f_ is the final diameter.

### 3.5. Oscillatory Rheometry

In the case of the parallel-plate rheometer, the complex viscosity (*η**) is affected by the gap and radius as follows:(19)$$\eta^{*}=\frac{\tau }{\dot{\gamma }}=\frac{2MH}{\pi \omega R^{4}}$$
where *H* is the gap, *M* is the torque, *ω* is the frequency, *τ* is the shear stress, and $\dot{\gamma }$ is the shear rate. 

Results from adjusting the data indicate that the protocol and corrections produced consistent data by reducing the test’s standard deviation by at least 52%, as seen in Figure 3.

Another problem solved with data treatment was the correction of the collapsing effect on the complex viscosity curves for all resins, as shown in Figure 4A at high frequencies. This was a major concern, since the expected trend in the complex viscosity was that the higher the temperature, the lower the viscosity. This implied that the effect of the temperature on the viscosity was violated. As the data were processed, introducing the swelling factor, the expected behavior of the complex viscosity was achieved. This correction can be observed in Figure 4B.

The correlations of solubility using PC-SAFT [26] and PR [21] equations of state were investigated using the results obtained by Arce and Aznar [27] and compared with the ones using the Henry constant and SLEOS in this study. The results using SLEOS to model the mixture of polymers and SCF were accurate. Other EOS were evaluated in order to identify differences between models.

For practicality, the solubility was compared at 458.15 K and both pressures (6.89 MPa and 13.78 MPa). Table 13 shows the comparison of solubility correlations using different equations.

The results show that SLEOS has a minor difference from estimations using PC-SAFT; however, differences from PR equation of state are significant, suggesting that PR may not be an appropriate model for PS+CO_2_ mixtures. On the other hand, SLEOS and PC-SAFT showed good agreement, so both can be considered as valid models to describe these systems.

## 4. Conclusions

In order to achieve a correct study of the viscoelastic behavior of polystyrene resins containing supercritical CO_2_ for foaming applications, it was necessary to develop an experimental procedure for a parallel-plate rheometer to assure repeatability and reproducibility.

Developed protocol controls variables from the instrument, the material and mixing phenomena. Control of material’s variables included mass and volume of the samples, deformations caused by thermal expansion and squeezing the polymer sample (for testing). Results indicate that the protocol offers consistent data by reducing standard deviations by at least 52% and by solving the problem of the collapsing effect of the complex viscosity at different temperatures (Figure 3). 

Estimations of swelling ratios were possible using SLEOS. In this model, the binary parameter played an important role for modeling the mixing phenomenon. This study shows the importance of using equations of state to adjust the dimensions of the samples for parallel-plate rheometry for these mixtures. 

## Figures and Tables

**Figure 1 polymers-14-03490-f001:**
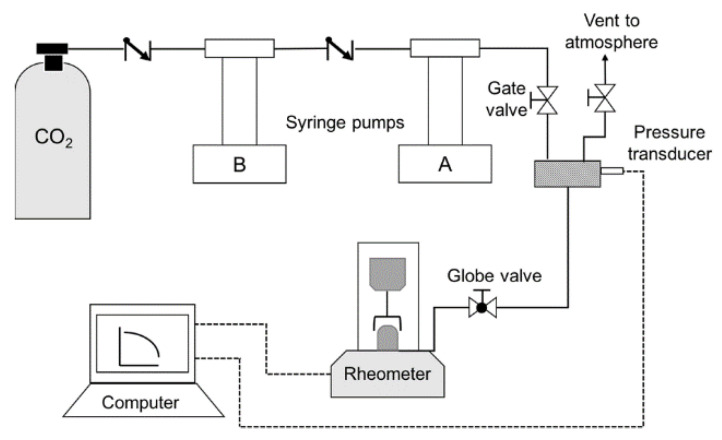
Schematic of the experimental setup: two syringe pumps used to pressurize with CO_2_ the cell of the rheometer already sealed and loaded with a PS sample.

**Figure 2 polymers-14-03490-f002:**
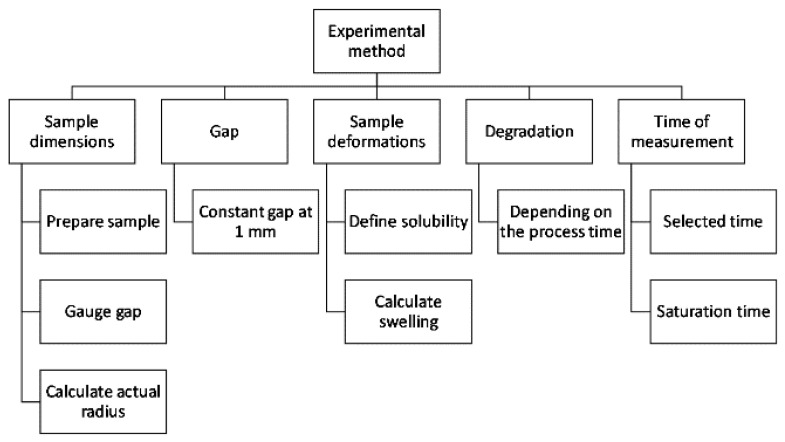
Experimental methodology for this study.

**Figure 3 polymers-14-03490-f003:**
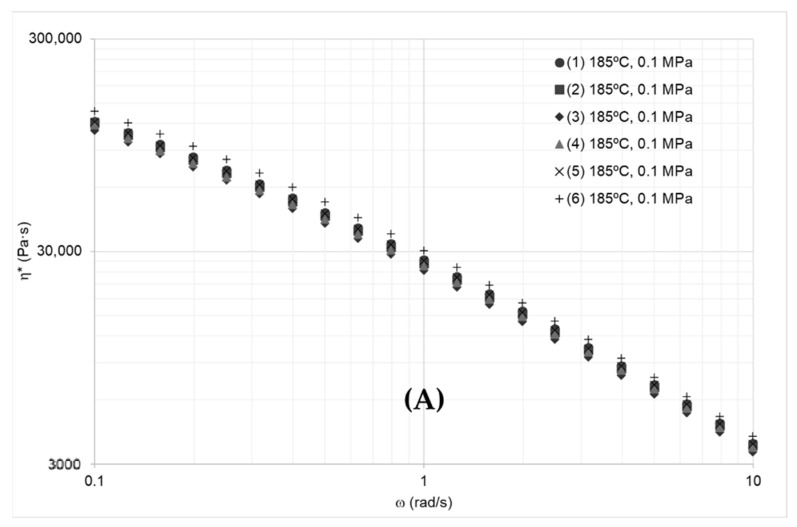
(**A**) Complex viscosity of several samples of Resin A as reported by the instrument (rheometer), (**B**) the same data recalculated using the actual radius in Equation (19).

**Figure 4 polymers-14-03490-f004:**
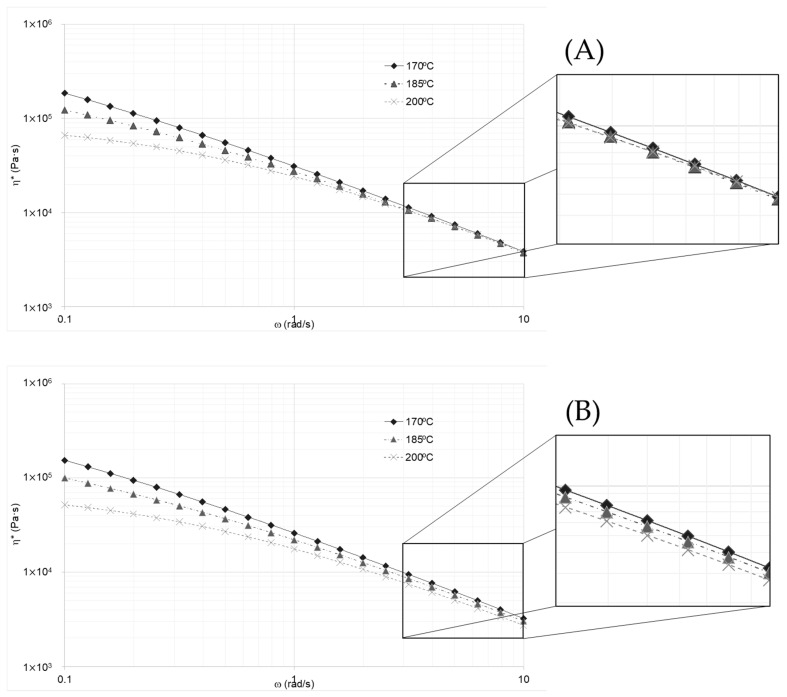
(**A**) Complex viscosity curves for resin A at different temperatures collapsing at high frequencies, (**B**) adjusted complex viscosity of resin A at different temperatures after correction for swelling, splitting off at high frequencies.

**Table 1 polymers-14-03490-t001:** Molecular weight distribution (MWD) parameters and the glass transition temperatures (Tg) for all the PS resins.

Resin ID	M_n_	M_w_	M_z_	Polydispersity	Peak M_w_	T_g_ (°C)
A	124,756	317,113	521,655	2.5	298,925	106.10
B	85,810	252,217	442,775	2.9	247,823	101.88
C	103,405	257,557	48,2252	2.5	224,748	106.34

**Table 2 polymers-14-03490-t002:** Constants used in Equation (15) to calculate the density of PS at atmospheric pressure.

*a* _0_	*a* _1_	*a* _2_	T_min_–T_max_	Ref.
1.0865	−6.19 × 10^−4^	1.36 × 10^−7^	100–200 °C	[23]
1.067	−5.02 × 10^−4^	1.35 × 10^−7^	79–320 °C	[24]

**Table 3 polymers-14-03490-t003:** Constants used in Equations (16) and (17) to calculate density of PS as a function of temperature and pressure.

*b* _0_	*b* _1_	C	T_min_–T_max_ (°C)	P_min_–P_max_ (bar)
2435	0.00414	0.09	115–249	0–2000
2521	0.00408	0.09	79–320	0–1800

**Table 4 polymers-14-03490-t004:** Calculated values of volumetric thermal expansion coefficient (*β_T_*) for PS.

*T*_0_ (°C)	*T_F_* (°C)	*V*_0_ (mm^3^)	*V_F_* (mm^3^)	*β_T_* (°C^−1^)
25	185	176.84	206.08	1.03 × 10^−3^
25	185	178.69	247.45	2.40 × 10^−3^
25	185	177.95	212.88	1.23 × 10^−3^
25	200	170.48	247.73	2.59 × 10^−3^
25	200	207.00	277.74	1.95 × 10^−3^

**Table 5 polymers-14-03490-t005:** Solubilities for CO_2_ in PS at various temperatures and pressures.

Pressure (MPa)	Temperature (K)	Solubility (g CO_2_/ g PS)
6.89	443.15	0.027
	458.15	0.025
	473.15	0.023
13.78	443.15	0.053
	458.15	0.049
	473.15	0.046

**Table 6 polymers-14-03490-t006:** Predicted binary interaction parameters for PS + CO_2_ at different temperatures using Equation (9).

Temperature (K)	k_12_
443.15	−0.144
458.15	−0.160
473.15	−0.180

**Table 7 polymers-14-03490-t007:** Characteristic parameters of PS and CO_2_ used in SLEOS.

Compound	ρ* (kg/m^3^)	*P** (MPa)	*T** (K)
PS	1108	387	739.9
CO_2_	1580	720.3	208.9 + 0.459 T − 7.56 × 10^−4^ T^2^

**Table 8 polymers-14-03490-t008:** Critical properties of the CO_2_ obtained by PR EOS.

T_C_ (K)	P_C_ (atm)	ω_a_
304.2	72.9	0.224

**Table 9 polymers-14-03490-t009:** Densities of CO_2_ at different temperatures and pressures obtained by PR EOS.

Pressure (MPa)	Temperature (K)	Density, ρ (kg/m^3^)
6.89	443.15	90.25
	458.15	86.14
	473.15	82.46
13.78	443.15	193.68
	458.15	182.52
	473.15	172.93

**Table 10 polymers-14-03490-t010:** Densities of PS calculated at different temperatures and pressures by using Equations (15–17).

P (MPa)	T (°C)	Average (Models)
		ρ (kg/m^3^)
0.1	170	985.38
	185	977.69
	200	970.07
6.89	170	990.20
	185	982.78
	200	975.42
13.78	170	994.82
	185	987.63
	200	980.52

**Table 11 polymers-14-03490-t011:** Swelling ratios for PS + CO_2_ at various temperatures and pressures using SLEOS.

P (MPa)	T (K)	Swelling Ratio
6.89	443.15	1.034
	458.15	1.032
	473.15	1.035
13.78	443.15	1.068
	458.15	1.067
	473.15	1.067

**Table 12 polymers-14-03490-t012:** Polymer discs dimensions after and during testing with SC-CO_2_.

P (MPa)	T (K)	h_i_ (mm)	d_i_ (mm)	β_T_ (°K^−1^)	Swelling Ratio	H (mm)	d_f_ (mm)
6.89	443.15	1.56	14.60	1.8 × 10^−3^	1.034	1	20.82
	458.15		14.60		1.032		21.02
	473.15		13.40		1.035		19.53
13.78	443.15		14.60		1.068		21.17
	458.15		14.60		1.067		21.38
	473.15		13.40		1.067		19.82

**Table 13 polymers-14-03490-t013:** Comparison of solubility using different EOS: SLEOS, PC-SAFT and PR.

		SLEOS	PC-SAFT	PR	SLEOS vs. PC-SAFT	SLEOS vs. PR
P (MPa)	T (K)	Solubility (g CO_2_/ g PS)			% difference	
6.89	458.15	0.025	0.026	0.030	5%	18%
13.78	458.15	0.049	0.054	0.059	9%	18%

## Data Availability

The data presented in this study are available on request from the corresponding author.

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
