# Peer review of "Estimation of the Effects of CO2 and Temperature on the Swelling of PS-CO2 Mixtures at Supercritical Conditions on Rheological Testing"

_polymers, 2022, doi:10.3390/polym14173490_

Round 1

Reviewer 1 Report

The manuscript deals with novel idea of developed a testing methodology to provide accuracy and repeability on  rheology testing using a parallel-plate rheometer in a pressure cell, based on the phenomena thermodynamically explained by the Sanchez-Lacombe Equation of State (SLEOS). in spite the idea is novel, however, it needs the following major corrections to be suitable for final publication.

1-English language should be revised carefully all over the manuscript to avoid the grammatical errors.

2- The abstract section should be rewritten to include some results.

3- some essential characterization for the optimum prepared polystyrene resin should be included such as XRD and SEM to proof its structure and morphology

Author Response

We thank the reviewers for their time reading our manuscript, their useful comments, and suggestions for improvements. Below are our itemised responses to referee 1, outlining changes made during the revision of the manuscript. These changes are also visually available through track changes mode in the text to facilitate version
comparison.

The following colour scheme has been used: (i) Reviewer 1 comment in
blue and (ii) authors' response in Black Bolded font.

The manuscript deals with novel idea of developed a testing methodology to provide accuracy and repeability on rheology testing using a parallel-plate rheometer in a pressure cell, based on the phenomena thermodynamically explained by the Sanchez-Lacombe Equation of State (SLEOS). in spite the idea is novel, however, it needs the following major corrections to be suitable for final publication.

1-English language should be revised carefully all over the manuscript to avoid the grammatical errors.
The newest version of the text was checked by native speaker
2- The abstract section should be rewritten to include some results.
Thank you for pointing out deficiencies in the abstract. We have now improved it by including the main results along with some lines expressing the benefits of our contribution. The
abstract now read as:
The use of supercritical CO2 as a blowing agent for polymeric foams instead of traditional blowing agents has been a trend in recent years. To achieve the final desired final properties of the polymeric foams, the rheological behavior of the material needs to be reliable. The polymer swelling in the samples for rheological testing affects the results of the viscoelastic properties of the material. This study proposes a new testing methodology to control the accuracy and repeatability of the rheological characterization for PS-SCO2 samples. To develop this methodology, three polystyrene resins with
different molecular weight distributions were studied at three temperatures (170, 185 and 200ºC) and
three pressures (0.1 MPa, 6.89 MPa and 13.78 MPa). The CO2 concentration was estimated and used in
the Sanchez-Lacombe Equation of State (SLEOS) to determine the polymer swelling, as it affects the
dimensions of specimens tested in high-pressure rheometers. The correction factors provided a
consistent trend of the viscosity concerning temperature and a decrease of up to 50% in the standard
deviation. The results of this study are crucial for accurate measurement of their viscoelastic
properties by parallel plate rheometry.
3- Some essential characterization for the optimum prepared polystyrene resin should be
included such as XRD and SEM to proof its structure and morphology.

Although these observations are interesting, the structure and morphology are beyond the scope of this paper. In any case, we greatly thank you for the feedback and it could beconsidered for future studies where such properties become relevant for the testing methodology.

Reviewer 2 Report

This paper should be revised. There are some deficiencies in expression and writing.
1. Abstract: “The results of this study are crucial not only in the production of PS foams but also in the accurate measurement of their viscoelastic properties by parallel plate rheometry”. How do the results have an effect on production?

2. Lines 28,34,39,53,55,74,159,208,218,236.

3. Lines 47,107,136,147,270,271.

4. Table 1: The meaning of the parameters should be explained.

5. Figures 3 and 4, it is unclear.

6. The explanations of tables and figures are not enough.

Author Response

We thank the reviewers for their time reading our manuscript, their useful comments, and suggestions for improvements. Below are our itemised responses to referee 1, outlining changes made during the revision of the manuscript. These changes are also visually available through track changes mode in the text to facilitate version comparison. The following colour scheme has been used: (i) Reviewer 2 comment in blue and (ii) authors' response in Black Bolded font.

This paper should be revised. There are some deficiencies in expression and writing.

The newest version of the text was checked for grammatical and spelling errors.

  1. Abstract: “The results of this study are crucial not only in the production of PS foams but also in the accurate measurement of their viscoelastic properties by parallel plate rheometry”. How do the results have an effect on production?

We apologize for the confusion in the abstract mentioning the production of PS. In this study, as we don't focus on the production of the PS foams, we modify the abstract to be clearer regarding our contributions. The abstract now reads as changes are highlighted):

The use of supercritical CO2 as a blowing agent for polymeric foams instead of traditional blowing agents has been a trend in recent years. To achieve the final desired final properties of the polymeric foams, the rheological behavior of the material needs to be reliable. The polymer swelling in the samples for rheological testing affects the results of the viscoelastic properties of the material. This study proposes a new testing methodology to control the accuracy and repeatability of the rheological characterization for PS-SCO2 samples. To develop this methodology, three polystyrene resins with different molecular weight distributions were studied at three temperatures (170, 185 and 200ºC) and three pressures (0.1 MPa, 6.89 MPa and 13.78 MPa). The CO2 concentration was estimated and used in the Sanchez-Lacombe Equation of State (SLEOS) to determine the polymer swelling, as it affects the dimensions of specimens tested in high-pressure rheometers. The correction factors provided a consistent trend of the viscosity concerning temperature and a decrease of up to 50% in the standard deviation. The results of this study are crucial for accurate measurement of their viscoelastic properties by parallel plate rheometry.

  1. Lines 28,34,39,53,55,74,159,208,218,236.

All these lines have been improved and reviewed. 

  1. Lines 47,107,136,147,270,271.

All these lines have been improved and reviewed. 

  1. Table 1: The meaning of the parameters should be explained.

All the parameters of Table 1 have been explained in detail on the manuscript:

Table 1. Molecular weight distribution (MWD) parameters and glass transition temperatures (Tg) for PS resins.

Where Mn is the number average molecular weight, Mw is the average molecular weight, Mz third moment of the MWD and Mw/Mn the polydispersity index.

  1. Figures 3 and 4, it is unclear.

The description of figure 3 and 4 have been improved and developed. 

Figure 3. (a) Complex viscosity of several samples of Resin A such as reported by the instrument (rheometer), (b) Same data recalculated using the actual radius in equation 19.

Figure 4. (A) Complex viscosity curves for resin A at different temperatures collapsing at high frequencies, (B) adjusted complex viscosity of resin A at different temperatures after correction for swelling, splitting off at high frequencies

  1. The explanations of tables and figures are not enough.

All the explanations, captions and descriptions of the tables and figures have been improved.

Round 2

Reviewer 1 Report

the manuscipt was improved and the modified version is accepted at its current state

Author Response

We thank the reviewer for appreciating our effort in enhancing the manuscript. We owe this new version to its valuable remarks.

Reviewer 2 Report

Thanks to the authors for the revision. There are some points that can be improved.

1. Line 43: “later used by Han and Ma[17]”. It should be revised “later used by Han and Ma [17]”.

2. Line68: “glass transition temperatures (Tg) for PS resins”. It should be revised “glass transition temperatures (Tg) for PS resins”.

3. Figures 3 and 4, it is unclear. The symbols are small and the colors are so similar, so it is difficult to obtain the information.

Author Response

We thank the reviewers for their time reading our manuscript, their useful comments, and suggestions for improvements. Below are our itemised responses to referee 1, outlining changes made during the revision of the manuscript. These changes are also visually available through track changes mode in the text to facilitate version comparison. The following colour scheme has been used: (i) Reviewer 2 comment in blue and (ii) authors' response in Black Bolded font.

Thanks to the authors for the revision. There are some points that can be improved.

  1. Line 43: “later used by Han and Ma[17]”. It should be revised “later used by Han and Ma [17]”.

Thank you for pointing out deficiencies in the line. The line now read as:

The most used rheometer consists of a slit die rheometer first adapted by Han [16] . This configuration was later used by Han and Ma[17] , Royer et al. [18] and Lee et al. [3,19]

  1. Line68: “glass transition temperatures (Tg) for PS resins”. It should be revised “glass transition temperatures (Tg) for PS resins”.

Thank you for pointing out deficiencies in the line. The line now read as:

Table 1. Molecular weight distribution (MWD) parameters and the glass transition temperatures (Tg) for all the  PS resins.

  1. Figures 3 and 4, it is unclear. The symbols are small and the colors are so similar, so it is difficult to obtain the information.

Figures 3 and 4 have been improved, including more detailed results and changes in the axis to get a better view. The new figures are included here for the convenience of the reviewer:

Figure 3. (A) Complex viscosity of several samples of Resin A such as reported by the instrument (rheometer), (B) Same data recalculated using the actual radius in equation 19.

Figure 4. (A) Complex viscosity curves for resin A at different temperatures collapsing at high frequencies, (B) adjusted complex viscosity of resin A at different temperatures after correction for swelling, splitting off at high frequencies.
